# Factors of Having Difficulties Raising 3-Year-Old Children in Japan: Usefulness of Maternal and Child Health Information Accumulated by the Local Government

**DOI:** 10.3390/children8121084

**Published:** 2021-11-24

**Authors:** Kimiko Tagawa, Miwako Tsunematsu, Masayuki Kakehashi

**Affiliations:** Department of Health Informatics, Graduate School of Biomedical and Health Sciences, Hiroshima University, Hiroshima 734-8553, Japan; tsunematsu@hiroshima-u.ac.jp (M.T.); kakehashi@hiroshima-u.ac.jp (M.K.)

**Keywords:** having difficulties raising, maternal and child health information, parenting, children’s health condition, edinburgh postnatal depression scale (EPDS)

## Abstract

**Background:** Difficulties raising children may be associated with depressive tendencies and abuse by parents, for which maternal and child health information may be useful. We clarified factors related to difficulties in raising children at the time of the 3-year-old child health checkup. **Method:** This was a retrospective cohort study. We used maternal and child health information collected from the time of pregnancy notification until the 3-year-old child health checkup. The subjects were the parents of 507 children who were born and eligible for the 3-year-old child health checkup between September 2013 and October 2017. Logistic regression and ROC analyses were performed. The dependent variable was “having difficulties raising children at the 3-year-old health checkup”. **Result:** Eleven factors were clarified as risk factors. Three major factors among them were having difficulties raising children at the 18-month-old checkup (adOR, 6.3; 95%CI, 3.0–13.9), actions are at the child’s own pace and adult instructions are difficult to follow at the 18-month-old health checkup (adOR, 5.0; 95%CI, 1.3–25.4), and EPDS score ≥ 2 (adOR, 3.4; 95%CI, 1.5–8.1). The AUC of this predictive model was 0.86. At a cutoff value of 0.387, the sensitivity was 79.7% and the specificity was 77.6%. **Conclusion:** Having difficulties raising children at the 3-year-old health checkup has factors from the time of pregnancy and requires continued support. It was possible and useful to use maternal and child health information when screening high-risk parents.

## 1. Introduction

Japan has taken many measures against the declining birthrate in the wake of the so-called “1.57 shock” when the total fertility rate reached 1.57 in 1990. However, the declining birthrate did not improve and the number of births in 2019 fell below 900,000 two years earlier than expected, reaching 865,234 [1]. According to the 2015 Japanese National Fertility Survey, the ideal number of children for a couple is 2.32 and the planned number of children is 2.01, and the reason for this difference is that more than 20% of married couples whose wives are in their 30s answered “Can’t bear mentally/physically the burden of childrearing anymore” [2]. Based on this result, child-rearing involves psychological and physical burdens. Under these circumstances, in 2015, as a new plan in the national health movement “Healthy Parents and Children 21 (Second Phase)” in Japan, “Seamless healthcare for pregnant women and infants” was set as a fundamental measure, and “Support tailored to parents who have difficulties raising their children” was set as a prioritized measure [3]. Regarding difficulty in raising children, related concepts include “childcare anxiety”, “childcare stress”, “childcare burden”, and “childcare difficulty”, and it was reported that this leads to abuse and depressive tendencies of parents [4]. In addition, although the number of children is decreasing due to the declining birthrate, the number of child abuse consultations is increasing year by year [5]. Many of the perpetrators of abuse are parents, and child-rearing anxiety has been reported to be one of the factors. “Healthy Parents and Children 21 (Second Phase)” also demonstrated the importance of being involved in the early stages of pregnancy, with “measures to prevent child abuse from pregnancy” as a priority issue. As a mechanism to support parents and children currently residing in Japan, each local government issues a maternal and child health handbook after the notification of pregnancy. Thereafter, it is stipulated by law to subsidize the cost of the medical examinations for pregnant women and infants, and health checkups for 18-month-olds and 3-year-olds. The percentage of infant health examinations exceeds 95% nationwide [6], and information on maternal and child health is collected well; however, such information has not been fully utilized. Yamagata [7] stated that the utilization of data obtained from infant health check-ups should be considered as an important issue and the usefulness of its utilization should be examined. Furthermore, it is expected to be useful not only for reporting to the national government, but also for daily maternal and child health work, and the utilization of maternal and child health information was also mentioned as an issue for “Healthy Parents and Children 21 (Second Phase)”.

Childcare anxiety, which is associated with parents having difficulties raising children, has been shown to increase children’s BMI [8] and to be a risk factor for poor self-regulation [9]. In addition to these aspects, supporting parents having difficulties raising children is important when considering the prevention of the declining birthrate and abuse. There are also reports that parents with 3-year-olds have the most difficulty in raising children [10]. In addition, the last year of the national health checkup is for 3-year-old children. Therefore, it may be important to consider the factors that make it difficult to raise a 3-year-old child. Clarifying such factors is essential in order to provide support. In addition, utilizing the health information of mothers and children does not require new labor; thus, the burden on parents is small. Currently, there are few studies analyzing the factors of parents having difficulties raising their children utilizing only Japanese maternal and child health information. In recent years, the data including maternal and child health information has been used to analyze the factors that affect parents and children [11,12]. We considered maternal and child health information to be important and decided to examine its usefulness. Therefore, the purpose of this study was to clarify the factors related to having difficulties raising children (having difficulties, hereafter) at the time of the 3-year-old health checkup, and to examine the usefulness of the maternal and child health information that is currently collected.

## 2. Materials and Methods

### 2.1. Participants

Participants were parents of children who were born and registered as inhabitants of A Town, Hiroshima Prefecture, Japan, between September 2013 and October 2017. The exclusion criteria were those who were not registered residents at the time of the 3-year-old child’s health checkup or those who did not undergo the health checkup.

### 2.2. Study Design

This was a retrospective cohort study using maternal and child health information. This information was collected by the local government at the following eight points: at the time of pregnancy notification, pregnancy, from childbirth to newborn visit, newborn visit, infancy, health consultation for 4–5-month-olds, 18-month-old child health checkup, and 3-year-old child health checkup.

### 2.3. Measurements

As information at the time of pregnancy notification, we used that provided to the local government by the submitter of the pregnancy notification. The age, height, weight, family structure, smoking, drinking, pregnancy history, medical history, and concerns of pregnant women were collected. The age, smoking, and drinking information was also obtained for the spouse/partner. The information during pregnancy was the medical information submitted to Town A by the medical institution conducting the obstetric health examination. The information was determined by the obstetrician to indicate no problem, followup, detailed examination, and necessary treatment. Based on the results, they were divided into two groups, no problem and problematic (followup, detailed investigation, and required treatment). The information from birth to infant home visits (including visit programs for all families with babies) were from the birth contact form submitted by parents and the number of contacts for the infant home visit. A checklist for child support, an Edinburgh postnatal depression scale (EPDS), and a Japanese version of the Mother-to-Infant Bonding Scale (MIBS-J) were provided by the mother during her infant’s home visit. The local government also collected the height, weight, head circumference, chest circumference, presence of abnormalities (at the time of childbirth and one-month health checkup), childbirth method, nutrition, and child’s condition at the time of visit. Based on this information, the growth of the baby and the health of the mother were divided into three categories: no issue, observation, and consultation recommendation.

The checklist for child support was created jointly by the Kyushu University Hospital Department of Child Psychiatry and the Fukuoka City Public Health Center and evaluates the childcare environment, including maternal support, using the following nine items. Problems during pregnancy (Yes or No), Have you ever had a miscarriage, stillbirth, or lost a child in the first year? (Yes or No), Have you consulted with a counselor, psychosomatic medicine, or psychiatrist? (Yes or No), Talk to spouse (Yes, No, or No spouse), Talk to mother (Yes, No, or No mother), Talk to others (Yes or No), Financial anxiety (Yes or No), Satisfied with the current home and environment (Yes or No), Did your family or close friends get sick or die during pregnancy? (Yes or No), Do you sometimes not understand why your baby is crying? (Yes or No), and Do you ever want to hit your baby? (Yes or No).

The EPDS is a 10-item self-administered questionnaire developed by Cox J.L. in 1987 [13] for postpartum depression screening. Each item is scored on a 4-point scale (0–3). The Japanese version was translated by Okano, and its validity and reliability were confirmed. The optimal cutoff value for the first month after delivery is 8/9 [14,15]. A score of 9 or higher indicates a possibility of postpartum depression. In this study, we set the EPDS score cutoff value from 1 to 14 each as an independent variable.

The MIBS-J is a Japanese version of the MIBS developed by Marks M.N. It was translated by Yoshida et al., and its validity and reliability were confirmed. It consists of 10 items, with 4 points for each item. The higher the total score, the poorer the bonding between the mother and the infant [16]. The MIBS-J has a two-factor structure of lack of affection (LA) and anger and rejection (AR).

The information on infancy was the frequency of attendance at infant health consultations conducted by local governments, the number of consultations, the number of telephone consultations, and the number of nutrition consultations. The health consultation for children 4 to 5 months of age (at 4 months old) is a maternal and child health project sponsored by the local government and is conducted at the health center. The information at the time of health consultation for children 4 to 5 months of age (at 4 months old) was provided in the questionnaire answered by the parents, the measured values of the child (height, weight, head circumference, and chest circumference), and the judgment results (no issues, observation, or consultation recommendation) by the interviewed public health nurse. The 1-year 6-month (at 18 months old) and 3-year-old child health checkups are legally mandated checkups in Japan. At the target age, the local government will inform the guardian, and the child will undergo a group health checkup at the health center. The information at the time of the 18-month-old child health checkup was a questionnaire answered by the guardian, measured values of the child (height, weight, head circumference, and chest circumference), and urinalysis results (urine protein, urine sugar, and urine occult blood). Further information included pediatrician and dentist consultations, and interviews with dental hygienists, nutritionists, and public health nurses. The information at the time of the 3-year-old child health checkup was from the questionnaire answered by the guardian, including the person who filled out the form, family structure, having difficulties raising children, items about abuse, and attendance of the health checkup (Appendix A).

### 2.4. Main Outcome

Our main outcome was having difficulties raising 3-year-old children. Parents were asked “Do you have difficulties raising children?”, at both the 18-month-old and 3-year-old health checkups. “Always feeling” and “sometimes feeling” were considered “Having difficulties raising children”, and “not feeling” was considered “Not having difficulties raising children” in this analysis. We defined having difficulties raising children as “the difficulties faced while raising children that are caused by a variety of factors and evident in children or parents” in this study. It is also defined in “Healthy Parents and Children 21 (Second Phase)” [3].

### 2.5. Analysis Method

Descriptive statistics were performed using the participants’ characteristics. At the time of the 3-year-old child health checkup, participants were classified by having or not having difficulties raising children. Based on the Shapiro-Wilk test in which participants were divided according to having difficulties at the 3-year-old stage, normality was not found in the following items: maternal age, gestational age (weeks), weight at birth, EPDS score, days from delivery to EPDS, and MIBS-J. Therefore, the Mann-Whitney U test was performed. The χ^2^ test was used for infant‘s sex and EPDS ≥ 9, and Fisher’s exact test was performed for birth order and number of siblings at 3 years old.

Logistic regression analysis was performed with having difficulties at 3 years old (0: Not having 1: Having) as the dependent variable. For the independent variable selection, univariate regression analysis was performed with having difficulties at 3 years old, and those with a p value of 0.1 or less were adopted. Since there were missing values and the amount of data for each variable was different, independent variables were selected by forward selection using Nagelkerke’s R^2^. Independent variables were included if *p*-values were less than 0.05. In this study, we set the EPDS score cutoff value from 1 to 14 each as an independent variable. The variables for each EPDS cutoff value were entered into the model in the same way as the other independent variables. However, they were entered one at a time, so that they would not be entered at the same time. Finally, the model with *p*-values of < 0.05 for all independent variables and a maximum Nagelkerke’s R^2^ was adopted. The multicollinearity potentially leads to the wrong identification in the predictive model. Therefore the variance inflation factor (VIF) was calculated between the independent variables to confirm multicollinearity. The receiver operating characteristic (ROC) analysis and the Hosmer-Lemeshow test were performed to examine the fit of the model. Prediction performance used the receiver operating characteristic area under the curve (AUC), sensitivity, and specificity.

Correlation analysis was performed with having difficulties and abusive behavior using the φ coefficient at 18-month-old and 3-year-old health checkups. Statistical analysis was carried out using R version 4.1.0 (packages: Epi, car, psych, pROC, rcompanion, and ResourceSelection). *p*-values lower than 0.05 were considered significant.

### 2.6. Ethical Considerations

The data used in this study were collected by a local government as existing maternal and child health information. Personally identifiable names and dates of birth were not collected when retrieving the data. This study was conducted after ethical approval (E-909-1) by the University Epidemiological Ethics Committee.

## 3. Results

During that period, children of 640 parents reached the target age for a 3-year-old health checkup. One hundred and thirty-three participants were excluded from the analysis, because 117 were not registered residents at the time of the 3-year-old child health checkup, and 16 did not answer the questionnaire for the 3-year-old child health checkup. The demographic characteristics and background information of the 507 participants are shown in Table 1. Most variables had 15–20% of missing values. The largest proportion of missing values was 50.9% (*n* = 260) with Amount of bleeding for delivery. The average maternal age at birth in Japan in 2013 was 31.6 years and the birth age of mothers in this study was 31.1 (SD = 5.1) years, so they were almost the same. At the 18-month-old and 3-year-old stage, the percentages of parents having difficulties were 25.2% and 40.8% (excluding no answer) in this study, and 25.6% and 33.8% nationwide, respectively. The study area showed slightly higher proportion at the 3-year-old stage. In this study, the average duration from the time of pregnancy notification to the time of the 3-year-old child health checkup was 3.7 (SD = 0.4) years. The mean total number of family members living together at the 3-year-old checkup was 4.0 (SD = 0.9). In addition, 465 mothers (91.7%) lived with their children’s fathers.

Univariate logistic regression analysis with having difficulties at the 3-year-old stage as the dependent variable selected 142 variables with a p value of less than 0.1. Parental smoking and drinking did not satisfy significance levels both at the time of pregnancy notification and at the 18-month-old health checkup. Having difficulties at the 18-month-old checkup (adjusted odds ratio [adOR], 6.3; 95%CI, 3.0–13.9), actions are at the child’s own pace and adult instructions are difficult to follow at the 18-month-old health checkup (adOR, 5.0; 95%CI, 1.3–25.5), and EPDS score ≥ 2 (adOR, 3.4; 95%CI, 1.5–8.1) were factors related to having difficulties at the 3-year-old stage. The other eight factors are shown in Table 2.

The logistic regression model is:$$log\left(\frac{p}{1-p}\right)=-2.30+1.23\;X01-1.11\;X02-0.91\;X03+1.32\;X04+1.22\;X05$$
+0.84 X06+1.28 X07+1.84 X08+1.71 X09+1.53 X10+1.62 X11

X1 to X11 show the variables in Table 2.

The AUC was 0.86. The *p* value of the logistic regression model using the obtained cutoff value was 0.95 based on the Hosmer-Lemeshow test. The VIF between the independent variables was 1.2 or lower. When the cutoff value was set to 0.387, the sensitivity, the specificity, the positive predictive value (PPV), and the negative predictive value (NPV) obtained in this study are shown in Figure 1 and Table 3. PPV and NPV depend on the prevalence. Therefore, PPV and NPV were calculated using the proportion of having difficulties raising children at the 3-year-olds stage in Japan in 2014 and 2017 and the sensitivity and specificity in this study (Table 3).

The φ coefficients of having difficulties raising children and abusive behavior were 0.22 (*p* < 0.001) and 0.21 (*p* < 0.001) at the 3-year-old and 18-month-old checkups, respectively. Furthermore, the φ coefficient of abusive behavior at the 18-month-old and 3-year-old checkups was 0.41 (*p* < 0.001).

## 4. Discussion

The purpose of this study was to clarify the factors of parents having difficulties raising their children at the 3-year-old health checkup. Furthermore, the usefulness of health information was examined. As a result, 11 factors related to parents having difficulties were clarified. These factors have been shown to be able to predict having difficulties at the 3-year-old stage and are consisted of variables that are maternal and child health information from pregnancy to the time of the 18-month-old health checkup. Having difficulties at the 3-year-old stage was associated with EPDS scores (X05 in Table 2), parents’ feelings of childcare anxiety, childcare stress, lack of confidence in childcare, and the behavioral characteristics of children (X11). This was consistent with preceding research on the associated factors with having difficulties and their consequences [4].

Among the 11 influential factors, “Illnesses since birth at the 4-month-old checkup (X07)” and “Frequent diarrhea at the 18-month-old checkup (X09)” were variables related to the children’s health condition. When a child becomes ill, parents have more work, such as going to the hospital and dealing with children’s symptoms, in addition to taking care of the child as usual. It is reasonable that a child’s health condition has a direct influence on having difficulties. It is also noted that there was a significant association between child health problems and postpartum depression [17]. Furthermore, there was a report on the specific relationship between a mother’s depression and the child’s diarrhea [18]. Infant health is also one of the factors related to achieving the role of the mother [19]. Because the factors that lead to child-rearing confidence are consistent with the factors related to the sense of achievement of the role of the mother [20], a child’s health status is also associated with child-rearing confidence and child-rearing anxiety. Thus, it is possible that “Frequent diarrhea at the 18-month-old checkup (X09)” influenced having difficulties indirectly through a mother’s depressive state, in addition to the direct pathway.

“Having difficulties raising children at the 18-month-old checkup (X08)” and “Feel frustrated “Not sure” at health consultation for 4-month-olds (X06)” are variables that parents themselves answered about their feelings. “Plan to return to work after maternity leave at the time of the newborn visit (at the newborn visit) (X03)” and “Mother’s overall judgment of ‘observation’ at the newborn visit (X04)” are variables about the mother. Therefore, these all were considered as parental factors. Before considering the reason for each influential factor, we must recognize general circumstances surrounding Japanese mothers. Japanese mothers are prone to childcare stress because they have few opportunities to come into contact with infants and their relationships with their neighbors may be weak [21]. We have to consider what is happening around the 4-month-old child stage to consider on X06. Four to five months of age is when children gradually adjust their breastfeeding or milk intervals and lifestyle rhythms, and when parents face new challenges such as the introduction of baby food. Therefore, the accumulation of child-rearing stress and the irritability of mothers can be detected as an influential factor. It must be noted that providing appropriate support during this period is suggested to have a large impact on resolving these problems. “Plan to return to work after maternity leave at newborn visit (X03)” was a negative factor. There are reports that working mothers have lower child-rearing stress than full-time housewives [22,23], and mothers who are planning to return to work are likely to be actually working at the 3-year-old health checkup. Thus, mothers who replied their plan to return to work are considered to have less rearing stress and not having difficulties. In addition, as to X04 and X08, we notice the condition of the mother in the early postpartum period was related to the subsequent child-rearing. This suggests the need for support from the early postpartum period.

At the 18-month-old checkup, “Actions are at the child’s own pace and adult instructions are difficult to follow (X11)” and “Fell and received medical treatment by 18 months old (X10)” are considered to be characteristics of children. Injuries, such as falls, may be related to their hyperactivity and restlessness. Children’s characteristics, such as hyperactivity, inattention, and impulsivity, are thought to be factors related to the feeling of difficulty in childrearing. Mothers of children who have characteristics, such as hyperactivity, find it difficult to “respond to behavior”. They therefore have negative feelings towards children and childcare [24,25]. Moreover, child-rearing stress increases when mothers judge their child’s temperament to be difficult [26], and parents of children with neurodevelopmental disorders have higher levels of parenting stress and future anxiety [27,28]. On the other hand, “adult’s instructions are difficult to follow” can be regarded as normal development for 18-month-old children. It may be necessary to understand not only whether there is a problem with the child’s behavior but also how the parent perceives the child. It was previously reported that 67.8% of children at 18 months had some unintended accident [29]. The causes of child accidents are diverse, including child problems, family background, family routines, environmental problems, and product problems [30,31]. Moreover, there is a misunderstanding that parents do not supervise [30]. The background may be the hyperactivity or impulsivity of the child, or a child without such characteristics may have an accident, and the parent may lose confidence in childcare. As a result, both cases are related to having difficulties.

“The child has older sisters (X02)” significantly reduced having difficulties at the 3-year-old stage despite the fact that the infant’s sex and the birth order of the child were not detected to be influential factors in this study. This seems at odds with other reports, because it is usually considered more difficult to raise boy children. Actually, boys have been reported to be at increased risk of violence, restlessness, and rebellious behavior [32]. There is also a study that boys were more spanked because of misbehavior [33]. In addition, it is also said that mothers find it difficult to raise boy children with inactive behavioral characteristics [30]. Moreover, it was also reported that mothers felt it difficult to raise their children if they had difficulties to raise the siblings [34]. This may conversely mean that mothers who have less difficulties to raising their older children then have less difficulty raising younger siblings. These considerations finally suggest that the experience of raising an older sister could give confidence to parenting and alleviate the difficulty of raising a child.

In this study, the EPDS score after delivery influenced the factors related to having difficulties at the 3-year-old stage. Therefore, the EPDS may be used as a tool to measure difficulties in raising children. The EPDS is currently used for the purpose of screening for postpartum depression with a cutoff value of 9/8 in Japan. In this study, the percentage of mothers who scored 9 points or higher was 9.7%, and the average value was 4.2 (median 4), which was almost the same as the reports in Japan [35]. Therefore, more than 90% of mothers were not considered to have postpartum depression when responding to EPDS. The EPDS measurement period for this study was 50.4 days after delivery (SD = 24.9), so the risk of postpartum depression cannot be mentioned, but it did affect having difficulties at the 3-year-old stage. Mothers with postpartum depression symptoms had child-rearing anxiety and exhaustion even when their children were five years old, and their children often behaved in a worrisome manner [36]. The EPDS also includes items that indicate depression, and studies that analyzed factor structure identified factors related to anxiety and depression [37,38]. As the EPDS predicted child-rearing anxiety at the 3-year-old checkup, it may lead to difficulties raising children. In the future, the EPDS can be administered to all postnatal mothers, not just those at risk of postpartum depression. In the USA, the EPDS is recommended to be performed one month, two months, four months, and one year after delivery [39]. As the results of the EPDS affect not only the time of the next EPDS but also the subsequent childcare, it is necessary to consider the timing and frequency of implementation, and its usage other than screening for postpartum depression.

In this study, neither the overall score nor the subscale of MIBS-J was an influential factor on having difficulties at the 3-year-old stage. In Japan, the MIBS-J is used to screen for bonding disorders. Bonding disorders are an important issue for perinatal mental health. As a result of classifying participants according to having difficulties at the 3-year-old health checkup, the MIBS-J score was significantly higher in the “having difficulties raising children” group. The relationship between postpartum depression and maternal bonding disorders was previously reported [40,41,42], and the EPDS scores at 1 month and 6 months after delivery were associated with “bonding” 1 year after childbirth [38]. In addition, a negative feeling towards pregnancy predicted postpartum depression and bonding disorders, and the effects of that feeling on postpartum depression depend on “bonding” [43]. Negative thoughts during pregnancy were not investigated in this study, nor was the diagnosis of postpartum depression. However, as the EPDS predicts having difficulties at the 3-year-old, “bonding” may be a factor for the timing of the MIBS-J. It is necessary to consider bonding disorders, such as using the MIBS-J, when visiting a newborn and at the subsequent infant health checkup. In addition, the cutoff value for bonding disorders and the effects of the presence of bonding disorders on difficulties raising children should be considered.

In this study, the items at the time of the 18-month-old health checkup affected having difficulties raising children at the 3-year-old checkup. Based on this, items were extracted at each time point, suggesting the usefulness of utilizing maternal and child health information in selecting predictors. PPV and NPV depend on the prevalence. Therefore, PPV and NPV were calculated using the proportion of having difficulties raising children at 3 year olds in Japan in 2014 and 2017 and the sensitivity and specificity in this study (Table 3). This model provided stable positive and negative predictive values even when the prevalence changed (Table 3); consequently, it can predict having difficulties raising children at the 3-year-old checkup.

Abuse behavior at 3-year-old checkup was weakly associated with having difficulties at the 3-year-old stage. In Japan, more than half of parents with preschool children regarded that “loud scolding” was discipline [44]. It may be helpful for the reduction in abuse to provide parents information on what is abuse behavior and to support parents who have difficulties raising children.

This study has several limitations. First, this was an analysis using maternal and child health information from one region of Japan. However, the average maternal age at birth and the frequencies of having difficulties at the 3-year-old stage were almost similar to nationwide data as described in the results section. However, the proportion of those who felt difficulties raising children at the 3-year-old checkup were known to exceed 35% in some prefectures including in Tokyo, Osaka, and Aichi [45]. Therefore, our results can be considered to represent more urban areas. Second, there was little information on smoking and economic conditions. Parental smoking was not a factor in this study. However, it has been reported to be associated with child behavioral problems and child development [46]. This study did not ask about previous smoking, and as the answer is self-managed, it is difficult to grasp the actual situation. Furthermore, no information, such as household income, was obtained, and the relationship with poverty has not been analyzed. In the future, a survey that takes these factors into consideration is required. Third, there was maternal and child health information from one local government. In the future, in order to create a generalizable forecast model, we need to increase the number of local governments.

## 5. Conclusions

We clarified that the outcome variable “Having difficulties raising children at the 3-year-old health checkup” had related 11 factors (i.e., “Having difficulties raising children at the 18-month-old checkup”, “Actions are at child’s own pace and adult instructions are difficult to follow at 18-month-old health checkup”, “EPDS score ≥ 2”, “Plan to return to work after maternity leave at the newborn visit”, etc.). Some factors had an effect from the time of pregnancy, suggesting the need for continuous support from an early stage. The maternal and child health information at the time of the 18-month-old health checkup from the time of pregnancy notification was useful to predict “having difficulties raising children” at the 3-year-old child health checkup.

## Figures and Tables

**Figure 1 children-08-01084-f001:**
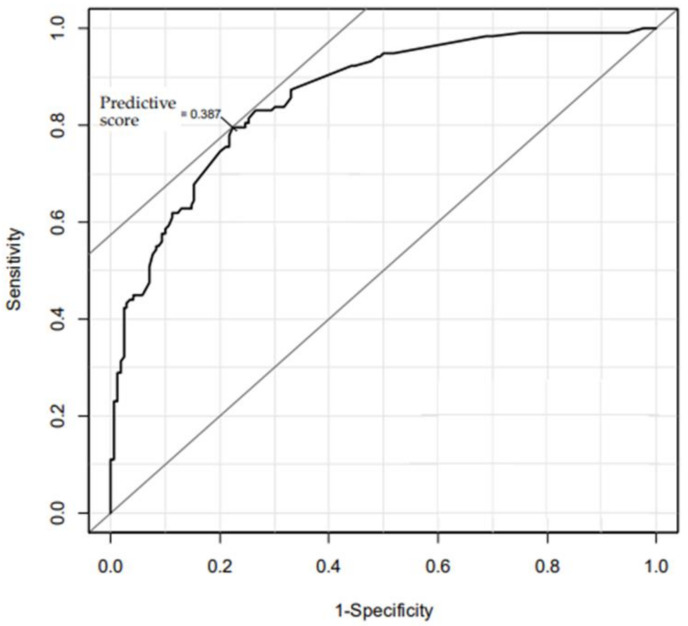
ROC curves for the predictive model of having difficulties in raising children by predictive score.

**Table 1 children-08-01084-t001:** Demographic characteristics and background information of those having difficulties raising children versus those not having difficulties raising children at 3-year-old health checkup.

	Total *N* = 507 (Maximum)		Having Difficulties Raising Children at 3-Year-Old Health Checkup *N* =207		Not Having Difficulties Raising Children at 3-Year-Old Health Checkup *N* = 300		
Variable	*N* (%)	Mean (SD)	*N* (%)	Mean (SD)	*N* (%)	Mean (SD)	*p*-Value
Maternal age ^#1^	506 (99.8)	31.1 (5.1)	207 (100)	31.6 (5.2)	299 (99.7)	30.8 (4.9)	0.27 ^†^
Infant’s sex							
Male	287 (56.6)		119 (57.5)		168 (56)		0.81 ^‡^
Female	220 (43.4)		88 (42.5)		132 (44)		
Birth order		1.6 (0.8)		1.5 (0.7)		1.7 (0.8)	0.14 ^§^
1	264 (52.1)		119 (57.5)		145 (48.3)		
2	185 (36.5)		68 (32.9)		117 (39)		
3	46 (9.1)		18 (8.7)		28 (9.3)		
4	9 (1.8)		1 (0.5)		8 (2.7)		
5	3 (0.6)		1 (0.5)		2 (0.7)		
Number of siblings at 3-year-old		1.0 (0.8)		0.9 (0.8)		1.0 (0.8)	0.45 ^§^
child health checkup							
0	129 (25.4)		61 (29.5)		68 (22.7)		
1	290 (57.2)		115 (55.6)		175 (58.3)		
2	71 (14.0)		25 (12.1)		46 (15.3)		
3	13 (2.6)		5 (2.4)		8 (2.7)		
≥4	4 (0.8)		1 (0.5)		3 (1.0)		
Gestational age (weeks) ^#^^2^	496 (97.8)	38.6 (1.8)	202 (97.6)	38.7 (1.9)	294 (98.0)	38.5 (1.7)	0.04 ^†^
Weight at birth	507 (100)	2953.1 (439.1)	207 (100)	2951.9 (442.0)	300 (100)	2954.0 (437.9)	0.88 ^†^
Days from delivery to	358 (70.6)	50.3 (25.0)	152 (73.4)	48.9 (23.0)	206 (68.7)	51.4 (26.3)	0.52 ^†^
EPDS conduct (days) ^#3^							
EPDS ^#^^4^	357 (70.4)	4.2 (3.4)	152 (73.4)	5.5 (3.8)	205 (68.3)	3.3 (2.6)	<0.001 ^†^
EPDS ≥ 9	35 (6.9)		26 (12.6)		9 (3.0)		<0.001 ^‡^
EPDS < 9	322 (63.5)		126 (60.9)		196 (65.3)		
MIBS-J ^#^^5^	355 (70.0)	1.7 (2.2)	151 (72.9)	2.3 (2.46)	204 (68.0)	1.2 (1.76)	<0.001 ^†^

The Mann-Whitney U test was used for group comparison of continuous variables. The χ2 test was used for group comparison of categorical variables. ^†^ Mann-Whitney U test, ^‡^ χ^2^ test, ^§^ Fisher’s exact test, Data with missing values were included. The numbers of missing values were: ^#1^ 1, ^#2^ 11, ^#3^ 149, ^#^^4^ 150, ^#^^5^ 152.

**Table 2 children-08-01084-t002:** Adjusted odds ratios (adORs) from the logistic regression model.

Independent Variables		adOR	95%CI	*p*-Value
X01	Problems during pregnancy	3.41	1.13–10.55	0.03
X02	The child has older sisters	0.33	0.14–0.71	0.006
X03	Plan to return to work after maternity leave at the newborn visit	0.4	0.21–0.76	0.006
X04	Mother’s overall judgement of “observation” at the newborn visit	3.73	1.15–12.95	0.031
X05	EPDS ≥ 2	3.39	1.52–8.13	0.004
X06	Feel frustrated “Not sure” at health consultation for 4-month-olds	2.32	1.21–4.50	0.011
X07	Illnesses since birth at health consultation for 4-month-olds	3.59	1.56–8.57	0.003
X08	Having difficulties raising children at 18-month-old health checkup	6.29	2.99–13.89	<0.001
X09	Frequent diarrhea at 18-month-old health checkup	5.53	2.17–14.95	<0.001
X10	Fell and received medical treatment by 18-month-old health checkup	4.63	1.6–14.53	0.006
X11	Actions are at child’s own pace and adult instructions are difficult to follow at 18-month-old health checkup	5.05	1.35–25.45	0.027
cutoff value		0.387		

CI; confidence interval Adjusted odds ratio shows relationship with “Having difficulties raising children (Having and Not having groups) at 3-year-old health checkup” and each factor using multiple logistic regression. Reference is set to none for all variables.

**Table 3 children-08-01084-t003:** Positive predictive value and negative predictive value by prevalence of having difficulties raising children at 3-year-old child health checkup.

	Predictive Score	Data at 3-Year-Old Health Checkup	
		2014 in Japan	2017 in Japan
Cutoff value	0.387		
Sensitivity	79.7	(79.7)	(79.7)
Specificity	77.6	(77.6)	(77.6)
Positive predictive value (PPV)	71.2	67.8	64.5
Negative predictive value (NPV)	84.6	86.6	88.2
Area under the curve	0.860		
Prevalence	41.1%	37.2%	33.8%

PPV and NPV by prevalence were calculated using the sensitivity (79.7) and specificity (77.6) of the analysis results. The prevalence at 3-year-old health checkup in 2014 and 2017 were data released by Japan.

## Data Availability

The data cannot be published, because they contain personal information, and we have not obtained consent to publish the data.

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
