# Peer review of "Factors of Having Difficulties Raising 3-Year-Old Children in Japan: Usefulness of Maternal and Child Health Information Accumulated by the Local Government"

_children, 2021, doi:10.3390/children8121084_

Round 1
Reviewer 1 Report
Methods and results should be better explained. L170-183: The authors describe different variable selection steps, involving some steps to manage missing data, and the Variance Inflation Factor (VIF). This paragraph should be improved by adding additional details. The exact sequence of steps should be clarified, the role of the VIF should be explained. Why did different p-values were used? Regarding EPDS variables, it is not clear what the Authors mean by "they were entered one at a time so that they would not be entered at the same time. Finally, an EPDS cutoff value that increased the value of Nagelkerke's R2 was adopted." The authors mention some steps that involve the creation of predictive models. To improve the reliability and reproducibility of the results, they should test the performance of the model on a test partition, which is left out from the fitting. Authors are referred to Bizzego et al. 2021, which describes a similar study. L194 - 215: It is not clear which of the variable selection steps produces the results in table 2. Why only 11 variables out of the 142 are shown? Are variables in Table 2 ranked by importance? Table 2 can be improved by adding the acronyms of the surveys/questionnaires that include the variables I think there is some missing text between L 215 and the Formula (L217-220). I think the Formula is also mathematically incorrect, or it should be better formatted. I see some discrepancies between the numbers reported in the text and those reported in Table 3.
Bizzego A, Gabrieli G, Bornstein MH, Deater-Deckard K, Lansford JE, Bradley RH, Costa M, Esposito G. Predictors of Contemporary under-5 Child Mortality in Low- and Middle-Income Countries: A Machine Learning Approach. Int J Environ Res Public Health. 2021 Feb 1;18(3):1315. doi: 10.3390/ijerph18031315. PMID: 33535688; PMCID: PMC7908191.
Author Response
We are grateful for the time and energy you have spent. Our responses to the reviewer’s comments are as follow:
Comment #1
L170-183: The authors describe different variable selection steps, involving some steps to manage missing data, and the Variance Inflation Factor (VIF). This paragraph should be improved by adding additional details. The exact sequence of steps should be clarified, the role of the VIF should be explained.
Response
The manuscript has been revised as follows on the role of the VIF: The multicollinearity is potentially leads to the wrong identification in the predictive model. Therefore the variance inflation factor (VIF) was calculated between the independent variables to confirm multicollinearity. (page 4 lines179-182)
Comment #2
Why did different p-values were used?
Response
We adopted p <0.1 for univariate regression analysis only. We performed univariate regression analysis to select the independent variables. At that time we relaxed the p value (p <0.1). That was because we referred to the following describe.
Sandro Sperandei. Understanding logistic regression analysis. Biochemia Medica.2014;24(1)12-18.doi: 10.11613/BM.2014.003.
Comment #3
Regarding EPDS variables, it is not clear what the Authors mean by "they were entered one at a time so that they would not be entered at the same time. Finally, an EPDS cutoff value that increased the value of Nagelkerke's R2 was adopted."
Response
We would like to add a point that our explanation was lacking.
We created 14 EPDS variables with different cutoff values. Each of them is considered as a candidate variable to be used in the regression model under the condition that only one EPDS variable is allowed to be used in the model. The logistic regression analysis was performed in the same manner as other variables, and the variables were selected by based on Nagelkerke's R². The manuscript has been revised as follows: In this study, we set the EPDS score cutoff value from 1 to 14 each as an independent variable. (page 4 lines 176-177) Finally, the model with p-values of <0.05 for all independent variables and a maximum Nagelkerke's R² was adopted. (page 4 lines 179-182)
Comment #4
The authors mention some steps that involve the creation of predictive models. To improve the reliability and reproducibility of the results, they should test the performance of the model on a test partition, which is left out from the fitting. Authors are referred to Bizzego et al. 2021, which describes a similar study.
Response
Thank you for introducing that interesting article. We have decided not to change our current analysis method. We tried splitting the data into training and test data sets. The results for the training data were almost identical to the results we are showing. It is difficult, as you might agree to, to do so because of small sample size of our data compared to the describe. In addition, the other reviewers have commented that our analysis method is correct.
Comment #5
L194 - 215: It is not clear which of the variable selection steps produces the results in table 2. Why only 11 variables out of the 142 are shown?
Response
The logistic regression analysis was performed the variables were selected by based on Nagelkerke's R². Independent variables were included if p-values were less than 0.05. Finally, the model with p-values of <0.05 for all independent variables and a maximum Nagelkerke's R² was adopted. As a result, 11 variables were adopted from 142 independent variables.
Comment #6
Are variables in Table 2 ranked by importance? Table 2 can be improved by adding the acronyms of the surveys/questionnaires that include the variables
Response
The variables in Table 2 are shown in chronological order. We have some variables with the same survey / survey acronym, so we responded by assigning numbers X1 to X11 to each variable.
Comment #7
I think there is some missing text between L 215 and the Formula (L217-220). I think the Formula is also mathematically incorrect, or it should be better formatted.
Response
Lines 212-215 were notes in Table 1, but the format in Table 1 has changed. Corrected the format in Table 1. There was no explanation for the formula. The following was added to the paper. The logistic regression model is: (page 7 line 226)
Also, we have revised the formula. (page 7 lines 227-228)
Comment #8
I see some discrepancies between the numbers reported in the text and those reported in Table 3.
Response
We didn't have enough explanation so we would like to add it.
PPV and NPV are used to determine the usefulness of the test, but depend on the prevalence. To test whether this model is useful even with different prevalence, we used in 2014 and 2017 prevalence of having difficulties rising children at 3-year-olds across Japan. PPV and NPV in the text were calculated from the sensitivity and specificity of the prediction model in this study. The prevalence rates for 2014 and 2017 in were not the rates calculated from the maternal and child health information collected in this study, but were published data throughout Japan. We have added the following sentences in the revised manuscript and changed Table 3:
PPV and NPV depend on the prevalence. Therefore, PPV and NPV were calculated using the proportion of having difficulties raising children at 3-year-olds in Japan in 2014 and 2017, and the sensitivity and specificity in this study (Table 3). (page 7 lines 234-237) The prevalence at 3-year-old health checkup in 2014 and 2017 were data released by Japan. (page 7 Table 3)
Reviewer 2 Report
Overall this is an interesting manuscript with well presented data, which I hope to see carefully revised and resubmitted. The introduction and methods are clear but the results could be clearer and the discussion section needs significant revision. The whole manuscript needs proofreading for typing errors and hard to read sentances. In particular, the discussion needs rewriting with language support as it is very hard to follow the arguments made. The end of the discussion includes some notes that should not be there. They indicate that the authors should have included some recommendations for future research but did not. Please delete the last 3 lines of the discussion and include some recommendations. The final paragraph of the discussion is also lacking references, where supporting evidence would be helpful.
The statistical methods are correct, although it is more common to identify the independent variables that should be included in the model during the study design phase then include them all rather than cherry-picking factors based on the outcome of the test of difference and entering them one by one. This does not need changing in the manuscript but is worth the authors' considering for future work. However, it would be nice to see the authors explaining the methods they used a little more in the results. For example, what does the difference in the PPV and NPV scores in 2014 and 2017 tell us? Why might it matter?
Author Response
Thank you very much for providing important comments. We are thankful for the time and energy you expended. Our responses to the reviewer’s comments are as follow:
Comment #1
The introduction and methods are clear but the results could be clearer and the discussion section needs significant revision. The whole manuscript needs proofreading for typing errors and hard to read sentances. In particular, the discussion needs rewriting with language support as it is very hard to follow the arguments made.
Response
Thank you for pointing out. We corrected typing errors. We added it to the text of the revised manuscript.(page 11 lines 365-373)
Comment #2
The end of the discussion includes some notes that should not be there. They indicate that the authors should have included some recommendations for future research but did not. Please delete the last 3 lines of the discussion and include some recommendations. The final paragraph of the discussion is also lacking references, where supporting evidence would be helpful.
Response
We have addressed your comment and removed the last 3 lines of the discussion. We have added references 41 and 42 to the manuscript. (page 11 line 378 and line 381)
Comment #3
The statistical methods are correct, although it is more common to identify the independent variables that should be included in the model during the study design phase then include them all rather than cherry-picking factors based on the outcome of the test of difference and entering them one by one. This does not need changing in the manuscript but is worth the authors' considering for future work.
Response
We agree your opinion that it is more common to identify the independent variables that should be included in the model during the study design phase. We performed univariate regression analysis to select the independent variables, rather than determining the independent variable from the outcome of the test of difference. We did not use that method because we aimed to predict the model from all maternal and child health information available in this study.
Comment #4
However, it would be nice to see the authors explaining the methods they used a little more in the results. For example, what does the difference in the PPV and NPV scores in 2014 and 2017 tell us? Why might it matter?
Response
We didn't have enough explanation so we would like to add it.
PPV and NPV are used to determine the usefulness of the test, but depend on the prevalence. To test whether this model is useful even with different prevalence, we used in 2014 and 2017 prevalence of having difficulties rising children at 3-year-olds across Japan. PPV and NPV in the text were calculated from the sensitivity and specificity of the prediction model in this study. The prevalence rates for 2014 and 2017 in were not the rates calculated from the maternal and child health information collected in this study, but were published data throughout Japan. We have added the following sentences in the revised manuscript and changed Table 3:
PPV and NPV depend on the prevalence. Therefore, PPV and NPV were calculated using the proportion of having difficulties raising children at 3-year-olds in Japan in 2014 and 2017, and the sensitivity and specificity in this study (Table 3). (page 7 lines 234-237) The prevalence at 3-year-old health checkup in 2014 and 2017 were data released by Japan. (page 7 Table 3)